# Regulation of mRNA translation by a photoriboswitch

**Kelly A Rotstan[1†], Michael M Abdelsayed[2†], Luiz FM Passalacqua[1], Fabio Chizzolini[1], Kasireddy Sudarshan[3], A Richard Chamberlin[1,4], Jiří Míšek[1,3]\*, Andrej Luptak[1,2,4]\***

[1]Department of Pharmaceutical Sciences, University of California, Irvine, United States; [2]Department of Molecular Biology and Biochemistry, University of California, Irvine, United States; [3]Department of Organic Chemistry, Charles University, Prague, Czech Republic; [4]Department of Chemistry, University of California, Irvine, United States

**Abstract** Optogenetic tools have revolutionized the study of receptor-mediated processes, but such tools are lacking for RNA-controlled systems. In particular, light-activated regulatory RNAs are needed for spatiotemporal control of gene expression. To fill this gap, we used in vitro selection to isolate a novel riboswitch that selectively binds the *trans* isoform of a stiff-stilbene (amino-*t*SS)–a rapidly and reversibly photoisomerizing small molecule. Structural probing revealed that the RNA binds amino-*t*SS about 100-times stronger than the *cis* photoisoform (amino-*c*SS). In vitro and in vivo functional analysis showed that the riboswitch, termed Werewolf-1 (Were-1), inhibits translation of a downstream open reading frame when bound to amino-*t*SS. Photoisomerization of the ligand with a sub-millisecond pulse of light induced the protein expression. In contrast, amino-*c*SS supported protein expression, which was inhibited upon photoisomerization to amino-*t*SS. Reversible photoregulation of gene expression using a genetically encoded RNA will likely facilitate high-resolution spatiotemporal analysis of complex RNA processes.

**\*For correspondence:**
jirimisek1@gmail.com (JM);
aluptak@uci.edu (AL)

[†]These authors contributed equally to this work

**Competing interests:** The authors declare that no competing interests exist.

## Introduction

Optogenetic techniques have transformed the biomedical sciences by controlling biological events with high spatial and temporal resolution through triggering signal transduction pathways *via* light-sensing proteins (*Fenno et al., 2011*; *Möglich and Moffat, 2010*; *Motta-Mena et al., 2014*; *Cambridge et al., 2009*); however, there is a need for photoactive molecules that can quickly and reversibly regulate cellular events at the RNA level. Photolabile ligands have previously been used to regulate RNA activity, but their photo-uncaging reactions are irreversible and relatively slow, requiring long UV light exposures, and resulting in limited temporal control (*Dhamodharan et al., 2018*; *Wulffen et al., 2012*; *Walsh et al., 2014*; *Young et al., 2009*; *Chaulk and MacMillan, 1998*; *Lucas et al., 2017*). Similarly, aptamers that bind a blue-light photosensitive protein have recently been developed for light-dependent regulation of translation (*Weber et al., 2019*). However, this system requires the expression of a relatively large regulatory protein that inhibits translation of a downstream mRNA when exposed to light and turns translation on only when slowly dark-adapted and dissociated from the RNA, preventing fast activation of protein expression. In contrast, photoreversible molecules, such as azobenzenes, have been employed to create several examples of photosensitive RNAs. For example, a light-responsive ribozyme has shown reversible activity, (*Lee et al., 2007*) and aptamers that bind photo-isomerizable ligands have been identified (*Hayashi et al., 2009*; *Lotz et al., 2019*; *Young and Deiters, 2008*), but these RNAs have only been characterized in vitro.

Riboswitches, cellular ligand-dependent regulatory RNAs that modulate transcription and translation in bacteria, and splicing, translation, and RNA stability in eukaryotes (*Breaker, 2011*) represent an attractive platform for the development of photosensitive regulatory RNAs. Riboswitches typically consist of an aptamer domain, which bind cellular targets, such as cofactors and metabolites, and an expression platform, which regulates gene expression by forming structures that affect downstream events, such as ribosome binding (*Fuchs et al., 2007*; *Breaker, 2012*). Many riboswitches sense their ligands during transcription; therefore, ligand-dependent conformational changes occur on the time-scales of synthesis of these RNAs (*Greenleaf et al., 2008*; *Uhm et al., 2018*). Furthermore, in many instances the aptamer and the expression platform are distinct domains of a riboswitch, allowing molecular evolution or engineering to convert genomic riboswitches into synthetic ones through replacement of the metabolite-sensing domains with in-vitro–selected aptamers (*Etzel and Mörl, 2017*).

We sought to develop a synthetic riboswitch system that regulates gene expression by binding a photoreversible ligand. We chose stiff-stilbene as the molecular scaffold for the ligand design, because stilbenes are well-characterized and exhibit fast photoswitching between the *trans* and *cis* states (*Quick et al., 2014*; *Waldeck, 1991*; *Szymański et al., 2013a*). We used in vitro selection (*Tuerk and Gold, 1990*; *Ellington and Szostak, 1990*) to isolate a novel RNA that selectively binds only the *trans* isoform of amino stiff-stilbene (amino-*t*SS). Chemical probing identified amino-*t*SS–induced RNA structural changes in both the aptamer domain and a downstream expression platform derived from a bacterial riboswitch. In vitro and in vivo functional analysis showed that the riboswitch, termed Were-1, can induce or inhibit translation of a downstream open reading frame upon exposure to a sub-millisecond pulse of light, through reversible photoisomerization of the ligand. Our results demonstrate how a genetically encoded light-responsive RNA can reversibly regulate gene expression using light, providing a new optogenetic tool to broaden the analysis of complex RNA processes in living cells (*You and Jaffrey, 2015*; *Liang et al., 2011*; *Jäschke, 2012*; *Szymański et al., 2013b*).

## Results

To isolate a new aptamer fused to a functional expression platform, we constructed an RNA pool derived from a bacterial SAM-I riboswitch (*Winkler et al., 2003*) by replacing its ligand-binding domain with a 45-nucleotide random sequence, partially randomizing its anti-terminator and terminator hairpins, and retaining its translation initiation sequences (*Figure 1—figure supplement 1*). We synthesized a photoactive ligand – a *trans* stiff-stilbene with an amino-terminated linker designed to maintain good cell permeability (*Figure 1a*). The ligand was also designed to have a narrow window for photoregulation of both isomerizations in order to keep the rest of the visible spectrum available for the potential readouts through luminescent or fluorescent reporters. Amino-*t*SS was characterized using UV-Vis and NMR spectroscopy to confirm photoisomerization to the *cis* conformation at 342 nm and back to the *trans* conformation at 372 nm, and to ensure that both isoforms are stable on timescales relevant to pulsed gene expression (*Figure 1—figure supplement 2*, *Figure 1—figure supplement 3*, *Figure 1—figure supplement 4*). The RNA pool- was selected in vitro to bind amino-*t*SS coupled to carboxylate agarose beads and eluted under denaturing conditions (*Ellington and Szostak, 1990*). We hypothesized that a pool of amino-*t*SS–binding aptamers would include motifs that do not bind the *cis* photoisoform of the ligand and that the *t*SS–binding conformation stabilizes the expression platform in a single state that affects either transcription or translation of a downstream open reading frame (ORF). After six rounds of in vitro selection, we cloned the pool into bacterial plasmids and tested individual sequences for amino-*t*SS binding by monitoring RNA–dependent changes in the fluorescence of the amino-*t*SS. One sequence (*Figure 1c*) showed markedly increased fluorescence of amino-*t*SS (*Figure 1—figure supplement 3*). This sequence, termed Werewolf-1 (Were-1) for its potential light-dependent conformational changes, was chosen for further analysis.

To assess the ligand-dependent structural modulation of Were-1, we performed multiple RNA structure-probing experiments, including digestions with T1 and S1 nucleases (*Peng et al., 2012*; *Reynolds and Gottesfeld, 1985*), terbium (III) footprinting (*Harris et al., 2014*), in-line probing (*Regulski and Breaker, 2008*), and selective 2' hydroxyl acylation by primer extension (SHAPE) (*Wilkinson et al., 2006*) using a range of amino-*t*SS concentrations (*Figure 1b* and *Figure 1—figure*

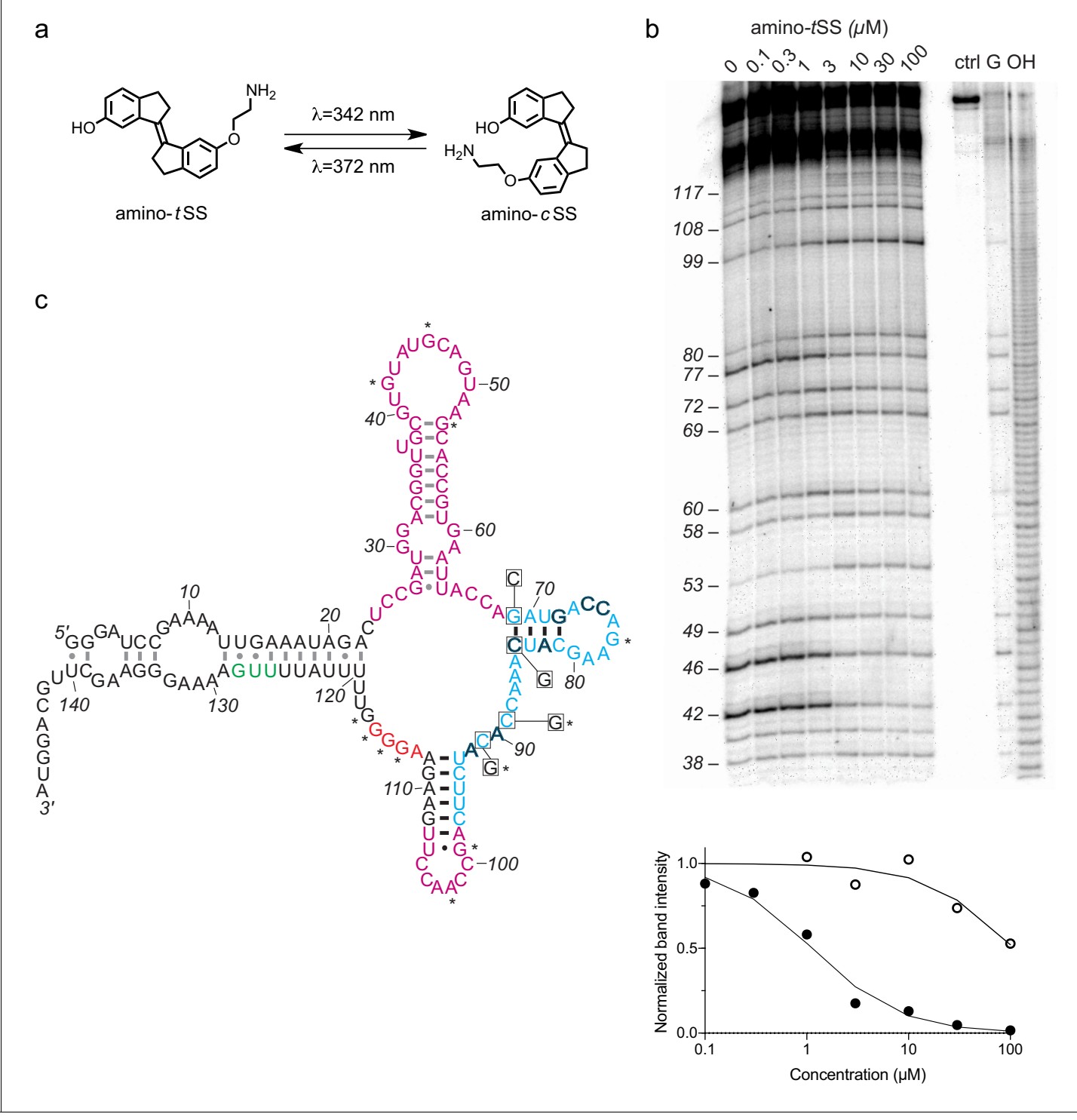

**Figure 1.** An amino-*t*SS-responsive aptamer. (a) Amino-*t*SS isomerizes from *trans* to *cis* conformation when exposed to 342 nm light, and back to the *trans* isoform at 372 nm. (b) RNase T1 probing of Were-1 structure. Right lanes contain a control with undigested RNA (ctrl), a T1-digested sequencing control (G), and a hydroxide-mediated partial digestion ladder (OH) of the RNA. The left lanes show partial T1 digestion in the presence of increasing amino-*t*SS at concentrations indicated above the gel image. The probing shows clear ligand-dependent changes—both increases (e.g. G53, G99, G114-117) and decreases (e.g. G 42, (G46, G77)—interspersed throughout the sequence. Below, an apparent $K_D$ of 1.1 µM was calculated based on the change in band intensity with increasing amino-*t*SS (dark, filled circles) for nucleotide G46, normalized to a control band (G72). Additionally, a $K_D$ of 108 µM was calculated based on the change in band intensity with increasing amino-*c*SS (open circles) for the same nucleotide and control (*Figure 1— figure supplement 4g*), suggesting high specificity for amino-*t*SS. An average $K_D$ value of 1.5 µM amino-*t*SS was calculated for changes in nucleotides

*Figure 1 continued on next page*

*Figure 1 continued*

G42, G46, G77, and G80. (c) Secondary structure prediction of Were-1 derived from all structural probing data in absence of the ligand (see also
***Figure 1—figure supplement 6***, ***Figure 1—figure supplement 7***, ***Figure 1—figure supplement 8***, ***Figure 1—figure supplement 9***). Partially
randomized regions (light blue), the Shine-Dalgarno sequence (red), the start codon (green), and the 3' terminus sequence are derived from the *B.
subtilis mswA* SAM-I riboswitch. The 5' part of the aptamer and the loop sequence (A98–U107) of the expression platform (pink) were selected from
random regions of the starting pool (***Figure 1—figure supplement 1***). Outlined dark letters are positions where the selected sequence differs from the
*B. subtilis* riboswitch expression platform. Boxed positions were mutated to the indicated nucleotides to identify regions of structural and functional
importance. Black base-pairs indicate stems that do not change in the presence of amino-*t*SS, and asterisks (*) indicate nucleotide positions that do
change in the presence of amino-*t*SS.

The online version of this article includes the following source data and figure supplement(s) for figure 1:

**Source data 1.** Derivation of dissociation constants from T1 probing data.
**Figure supplement 1.** In vitro selection pool design for a photoriboswitch.
**Figure supplement 2.** UV absorption and $^1$H NMR spectroscopy of E/Z isomerization of amino-*t*SS and amino-*c*SS.
**Figure supplement 2—source data 1.** UV spectra of amino-*t*SS and amino-*c*SS.
**Figure supplement 3.** $^1$H and $^{13}$H NMR spectra of *t*SS, amino-*t*SS, and amino-*c*SS.
**Figure supplement 4.** Stability of amino-*t*SS and amino-*c*SS in water and DMSO analyzed by UV-vis spectroscopy.
**Figure supplement 5.** Fluorescence of Were-1 bound to amino-*t*SS.
**Figure supplement 5—source data 1.** Fluorescence spectrum of RNA–bound amino-*t*SS.
**Figure supplement 6.** Structural probing of Were-1 in presence of amino-*t*SS using SHAPE and S1 nuclease digestion.
**Figure supplement 6—source data 1.** Derivation of apparent Kd values from RNA probing data.
**Figure supplement 7.** Structural probing of Were-1 in presence of amino-*t*SS, amino-*c*SS, *t*DHS, *t*S, and SAM by terbium (III) footprinting.
**Figure supplement 8.** Structural probing of Were-1 in presence of *t*DHS, *t*S, and SAM by T1 nuclease digestion and in-line probing.
**Figure supplement 9.** Structural probing of Were-1 in presence of amino-*c*SS, *t*DHS, *t*S, and SAM by SHAPE, and amino-*c*SS titration by T1 nuclease
digestion.

*supplement 4*). The changes in the pattern of RNA probing suggested that Were-1 undergoes conformational modulation upon introduction of amino-*t*SS in both the sequence derived from the randomized region (the aptamer domain) and the expression platform (***Figure 1b*** and ***Figure 1—figure
supplement 7***, ***Figure 1—figure supplement 8***, ***Figure 1—figure supplement 9***). Control experiments with amino-*t*SS analogs, such as *trans*-stilbene (*t*S), 4,4-*trans*-dihydroxystilbene (*t*DHS), and S-adenosyl methionine (SAM), showed no change in the probing patterns (***Figure 1—figure supplement 7***, ***Figure 1—figure supplement 8***, ***Figure 1—figure supplement 9***), suggesting that Were-1 is specific for amino-*t*SS. Analysis of the T1 probing experiments revealed a $K_D$ of 1.1 µM, based on the change in band intensity with increasing amino-*t*SS for nucleotide G42, normalized to a control band G72 (***Figure 1b***). A 108 µM $K_D$ was calculated based on the same band intensity change with increasing amino-*c*SS (***Figure 1—figure supplement 9***), revealing a 100-fold specificity for the target ligand, amino-*t*SS. The average amino-*t*SS $K_D$ derived form the T1 nuclease probing (at positions 42, 46, 77, and 80; normalized to the band intensity at position 72) was of 1.5 µM, whereas the nuclease S1 probing revealed a $K_D$ of 0.4 µM (based on band intensity change for positions A44-G46), and the terbium (III) footprinting yielded a somewhat higher apparent $K_D$ of 4.8 µM (calculated based on the change in intensity at positions A113 and U107, normalized to G134 control band; ***Figure 1—figure supplement 7***).

To establish the location of the amino-*t*SS aptamer domain, we modeled the secondary structure of Were-1 based on the probing data and created mutants hypothesized to affect ligand binding affinity or RNA structural stability (***Figure 1c***), and tested them in vitro and in vivo. The secondary structure modeling did not support a conformation containing a Rho-independent transcriptional terminator, in part because the selected sequence contained two mutations (C90A and U92A), which are predicted to disrupt the stability of a full-length transcription-terminating helix (***Figure 1c***). The structure-probing experiments (***Figure 1b*** and ***Figure 1—figure supplement 6***, ***Figure 1—figure supplement 7***, ***Figure 1—figure supplement 8***, ***Figure 1—figure supplement 9***) supported a secondary structure consisting of four helical segments. Ligand-dependent changes mapped primarily to the loops of the molecule, suggesting that it does not undergo rearrangement in secondary structure upon amino-*t*SS binding. Instead, the structure probing experiments suggest that the loops either contact the ligand directly or make tertiary contacts to form its binding pocket.

To test whether Were-1 can directly couple light-induced states of the ligand to the activity of the expression platform in vitro, the aptamer was tested for amino-*t*SS–dependent conformational

changes using a strand-displacement assay that mimics mRNA binding by the bacterial ribosome (*Martini et al., 2016*; *Zhang and Winfree, 2009*; *Martini et al., 2015*). We designed a DNA duplex in which the longer strand has a toehold sequence corresponding to the reverse complement of the Shine-Dalgarno sequence of Were-1 (*Supplementary file 1* - Supplementary Table 1) and a fluorophore to assess whether the shorter DNA strand, containing a quencher chromophore, is displaced through RNA:DNA hybridization with Were-1 (*Figure 2—figure supplement 1a*). Polyacrylamide gel electrophoresis (PAGE)-purified Were-1 bound the toehold readily, but the strand displacement was diminished in the presence of amino-*t*SS (*Figure 2—figure supplement 1b*). Testing the toehold binding at various concentrations of amino-*t*SS revealed a dose dependence with a half–maximal inhibition of toehold binding at ~6 µM (*Figure 2—figure supplement 1c*).

We next asked whether the ribosome mimic binds this RNA during in vitro transcription (*Figure 2a* and *Figure 2—figure supplement 2*). In the absence of the ligand, the RNA bound the toehold efficiently, showing a robust increase of fluorescence immediately after transcription initiation. In contrast, the addition of high concentration (14.8 µM) of amino-*t*SS strongly abrogated new binding of the toehold, as revealed by almost a full reduction in the slope of the fluorescence expansion curve (*Figure 2b*). Intermediate concentrations of amino-*t*SS were tested to assess RNA binding and specificity, yielding a ligand-dependent response with a half-maximum of ~4 µM amino-*t*SS (*Figure 2c*).

Probing-derived secondary structure modeling of Were-1 suggested formation of a ligand-insensitive hairpin (positions 69–84). To confirm the presence of this structural element, we created a variant containing a single mutation (G69C), that was predicted to disrupt the stem, and a presumed compensatory mutant (G69C/C84G) (*Figure 1c*). The G69C variant showed diminished response to amino-*t*SS, whereas the G69C/C84G double-mutant exhibited partially restored activity, suggesting that these two positions are indeed part of a helix. Other variants, C89G and C91G, designed based on parts of the sequence that showed amino-*t*SS–dependent changes in the structure-probing experiments, both showed decreased sensitivity to the ligand, suggesting that they are essential for ligand binding. Furthermore, when testing toehold binding using the purified *cis* isoform of the stiff stilbene (amino-*c*SS), as well as other stilbenes, such as *t*S, *t*DHS, and *trans* stiff stilbene (*t*SS; Were-1 ligand lacking the aminolated linker), no significant changes in fluorescence were observed (*Figure 2c* and *Figure 2—figure supplement 3*). These results demonstrate that amino-*t*SS stabilizes the RNA in an 'OFF' (ribosome-inaccessible) conformation in a dose-dependent manner and with high ligand specificity.

In order to test whether the Were-1 RNA interaction is selective for amino-*t*SS, and potentially acts as an amino-*t*SS riboswitch, we created a construct consisting of the putative riboswitch, including its minor start codon from *Bacillus subtilis* that was present in the starting pool, followed by a firefly luciferase (Fluc) ORF lacking its endogenous start codon. We chose luciferase for the readout because, unlike fluorescent proteins, it does not require excitation light and quantitatively yields photons immediately after the enzyme is synthesized. Based on the toehold assays, we hypothesized that in absence of amino-*t*SS, Were-1 would be transcribed in a conformation promoting the translation of the luciferase enzyme, whereas the presence of amino-*t*SS would stabilize a conformation preventing efficient translation initiation, downregulating the luciferase expression (*Figure 2d*). Using a purified bacterial in vitro transcription/translation system, luciferase production was measured in the presence and absence of amino-*t*SS. In absence of the ligand, the construct exhibited robust luciferase production, demonstrating that the unbound aptamer promotes protein production from a downstream ORF (*Figure 2e*). In contrast, when amino-*t*SS was added, protein production decreased in a dose-dependent manner (*Figure 2f*). To confirm that the ligand itself did not impact the in vitro translation system, luminescence was tested with a control plasmid lacking the riboswitch, and no amino-*t*SS sensitivity was observed (*Figure 2—figure supplement 4*).

To further test the riboswitch, we incorporated the construct into a bacterial plasmid and induced its expression in *Escherichia coli* cells (*Figure 2g*). Bioluminescence, due to luciferase expression and activity, was robust in the absence of ligand, and when the cells were incubated in the presence of amino-*t*SS, bioluminescence was again diminished in a dose-dependent manner (*Figure 2h and i*). To confirm the specificity of Were-1 for amino-*t*SS, we incubated cells in the presence of amino-*c*SS, *t*S and *t*DHS, and observed no significant change in bioluminescence (*Figure 2i*). To determine the effect of an alternate start codon on Fluc expression, we changed the UUG start codon in the Were-1-Fluc plasmid to AUG. Bioluminescence was higher in the AUG samples compared to the wild-type

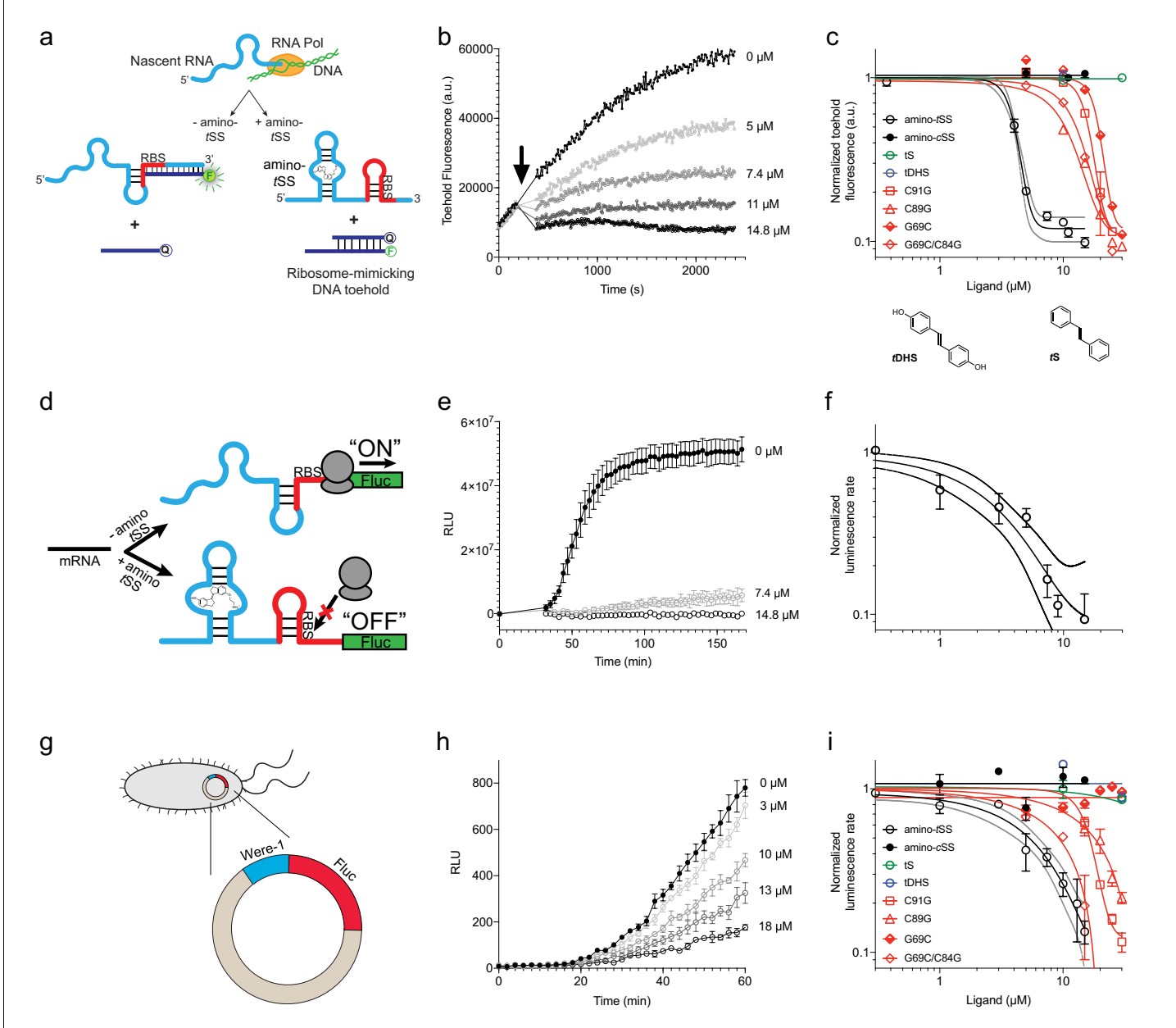

**Figure 2.** Translation regulation by the Were-1 riboswitch. (**a**) Schematic of co-transcriptional binding of Were-1 RNA to amino-*t*SS in the presence of a toehold-reporter complex. In absence of amino-*t*SS, the transcribed RNA exposes the ribosomal binding site (RBS), enabling binding of the complementary region of the toehold reporter, displacing the quencher strand, and producing a fluorescence signal. In presence of amino-*t*SS, the RNA binds the ligand, sequestering the RBS and preventing displacement of the quencher strand. (**b**) Co-transcriptional response of Were-1 to different concentrations of amino-*t*SS using the toehold reporter. Initial transcriptions of Were-1 without ligand show identical increase in toehold fluorescence for all samples. When amino-*t*SS is added (arrow), a dose-dependent decrease in fluorescence is observed. (**c**) Response (± SEM; n = 81) of Were-1 (black, open circles), and its variants (red) C89G (triangles), C91G (squares), G69C (half-shaded diamonds), and G69C/C84G (open diamonds), in the presence of amino-*t*SS shows a shift in dose-dependence for single mutations, particularly G69C, and partial recovery of activity for the G69C/C84G double mutant. Were-1 shows no response in the presence of amino-*c*SS (black circles), *trans*-stilbene (*t*S, green, open circles) and *trans*-4,4-dihydroxystilbene (*t*DHS, blue, open circles). Structures of *t*DHS and *t*S are shown below the graph. (**d**) Schematic of amino-*t*SS-dependent inhibition of protein expression in vitro using a Were-1-firefly luciferase (Were-1-Fluc) construct. In absence of the ligand, the RBS is exposed and luciferase is translated, whereas in presence of amino-*t*SS, the RBS is sequestered, abrogating Fluc expression. (**e**) In vitro translation of the Were-1-Fluc construct. Robust luminescence is observed when no ligand is present, but the signal is significantly lower in presence of amino-*t*SS. (**f**) Response (± SEM; n = 58) of the Were-1–regulated protein expression to amino-*t*SS. (**g**) Schematic of the Were-1-Fluc construct incorporated into a bacterial plasmid. (**h**) Were-1–controlled Fluc gene expression in *E. coli*. Bioluminescence is observed in absence of amino-*t*SS, and progressively diminished with increasing amino-

*Figure 2 continued on next page*

*Figure 2 continued*

*t*SS. (i) Expression of Were-1-Fluc (± SEM; n = 257) in vivo (black, open circles), and its variants (red) C89G (triangles), C91G (squares), and G69C/C84G (open diamonds), in the presence of amino-*t*SS, show a dose-dependent response. Were-1 mutant G69C (half-shaded diamond) and Were-1 in the presence of amino-*c*SS (black circles), *trans*-stilbene (green, open circles) and *trans*-4,4-dihydroxystilbene (blue, open circles) showed no change in bioluminescence, whereas the G69C/C84G double mutant shows restoration of activity similar to wild-type levels. Note, dose-response graphs (c, f, i) are on a log-log scale. The apparent amino-*t*SS $IC_{50}$s are 3.9 ± 0.2, 2.5 ± 1.0, and 5.3 ± 1.1 µM for the toehold (c), in vitro translation (f), and in vivo expression (i), respectively. Dashed lines correspond to the 95% confidence interval of the binding model.

The online version of this article includes the following source data and figure supplement(s) for figure 2:

**Source data 1.** Kinetic data for Were-1 activity.
**Figure supplement 1.** Binding PAGE-purified Were-1 RNA to a ribosome-mimic.
**Figure supplement 1—source data 1.** Time course of toehold binding by Were-1.
**Figure supplement 2.** Co-transcriptional binding of a ribosome-mimic in vitro under various $Mg^{2+}$ conditions.
**Figure supplement 2—source data 1.** Effect of Mg (II) on toehold binding by Were-1 in absence and presence of amino-*t*SS.
**Figure supplement 3.** Co-transcriptional Were-1 binding of a ribosome-mimic in vitro in the presence of *t*SS (in 10% DMSO).
**Figure supplement 3—source data 1.** Effect of *t*SS in Were-1 binding using the toehold assay.
**Figure supplement 4.** Translation of a control plasmid, Luc2-pET, lacking the Were-1 riboswitch is not inhibited in vitro by amino-*t*SS.
**Figure supplement 4—source data 1.** Lack of inhibition of Were-1-independent expression system by amino-tSS.
**Figure supplement 5.** The effect of the canonical start codon on Were-1-Fluc expression in vivo.
**Figure supplement 5—source data 1.** Comparison of protein expression by amino-tSS in Were-1 variants containing UUG or AUG start codon.

UUG construct and showed a similar amino-*t*SS–dependent response (*Figure 2—figure supplement 5*). Testing the above-described mutants confirmed the G69/C84 interaction, and the importance of the C89 and C91 positions for ligand binding. Taken together, our results demonstrate that Were-1 controls amino-*t*SS–dependent protein expression in vitro and in vivo, acting as a translational riboswitch.

We next asked whether Were-1 could regulate gene expression in a light-dependent manner, acting as a photoriboswitch. For this to occur, the *trans* isoform of the stiff stilbene must photoisomerize to its *cis* state, preventing binding of the Were-1 aptamer domain and promoting expression of a downstream ORF.

Riboswitches are sensitive to co-transcriptional events because they are capable of adopting different RNA conformations as they are transcribed by RNA polymerase (*Frieda and Block, 2012*). To first monitor changes in RNA folding over time, we used the toehold-fluorophore system to determine whether the DNA duplex was in a bound state (no strand displacement) versus an unbound state (strand displacement releasing the quencher DNA, yielding fluorescence) during transcription. Our results show that when the amino-*t*SS-bound Were-1 structure was irradiated at 342 nm of light, the toehold fluorescence increased, suggesting that the RNA increased the binding to the ribosome mimic present on the DNA toehold. Furthermore, when exciting amino-*c*SS at 372 nm of light to switch the ligand to its *trans* isoform, we observed a decrease in toehold fluorescence growth, indicating that the photo-generated amino-*t*SS was able to re-bind the Were-1 RNA. The irradiation was repeated until all toehold was bound, with each switch showing consistent results (*Figure 3—figure supplement 1a*). This experiment shows reversible, wavelength-dependent binding of the ribosome mimic, emulating light-dependent protein expression from a downstream open reading frame. Furthermore, analysis of the toehold binding as a function of exposure length revealed that even millisecond pulses of light at 340 ± 10 nm ($\Phi_q$ = 1.4*$10^{-2}$ W/cm$^2$) result in a detectable change in toehold fluorescence when compared to unexposed transcription reaction, suggesting that the system has the capacity to act on timescales that are faster than a single nucleotide addition during RNA synthesis (*Figure 3—figure supplement 1b*), which for *E. coli* is in tens of milliseconds range (at 37°C) (*Manor et al., 1969*; *Bremer and Yuan, 1968*).

To study the system further, we used the Were-1-luciferase construct and the purified bacterial in vitro transcription/translation system and tested whether luciferase expression could be regulated by our putative photoriboswitch. We found that when the reaction was irradiated at 342 nm of light, luminescence increased, suggesting that Were-1's conformation changed to expose the RBS, enabling luciferase expression. When the *cis* stilbene was photoisomerized to the *trans* state (amino-*t*SS) at 372 nm of light, luciferase protein production slightly decreased (*Figure 3—figure supplement 2*). These data are consistent with an in vitro activity of a photoriboswitch.

Finally, we used *E. coli* containing the Were-1-Fluc construct to determine whether gene expression can be regulated with a pulse of light. As shown in *Figure 2i*, bioluminescence was greatly diminished in the presence of amino-*t*SS. Upon exposing the bacteria to 342 nm light, we saw a modest increase in bioluminescence, relative to the unexposed control samples (*Figure 3a and b*, *Figure 3—figure supplement 3*), suggesting that upon photoisomerization of amino-*t*SS to the *cis* isoform with a pulse of light, Were-1 changed its conformation to expose its RBS and support luciferase production. Given that photoisomerization of stiff stilbenes from *trans* to *cis* is not quantitative (meaning, only about 40% of the ligand is photoisomerized; see *Figure 1—figure supplement 2b*), the trend in protein expression at individual amino-*t*SS concentrations can be rationalized in the following way: at low ligand concentrations, translation is weakly inhibited and photoisomerization has little impact on the rate of protein synthesis; at intermediate ligand concentrations (near the $IC_{50}$ concentration, that is, at the most concentration-sensitive segment of the ligand-response curve) photoisomerization leads to a change in amino-*t*SS concentration that results in the biggest change in translation initiation; and, at high ligand concentrations, photoisomerization results in little derepression of translation initiation because enough amino-*t*SS remains to continue inhibiting protein synthesis. Thus the incomplete photoisomerization of the stiff stilbene results in the bell-shaped ligand dependence upon light exposure. Additionally, in another experiment, we excited cells with a different wavelength of light (500 nm) that should not impact the isomerization of the ligand (*Figure 3—figure supplement 3*). Both controls showed lower bioluminescence compared to the 342 nm–excited samples, suggesting that the increased protein expression was due to derepression of translation initiation upon photoisomerization of amino-*t*SS to amino-*c*SS, although the amplitude of the effect is variable, perhaps due to variability of Were-1–regulated protein expression at different cell densities.

To further analyze the system, we tested the temporal dependence of Were-1–regulated Fluc production by exposing an amino-*t*SS–containing bacterial culture to 342 nm light for various lengths of time. Relative to unexposed controls at the same ligand concentration, the highest luciferase expression was seen at an exposure time of 500 μs (*Figure 3b*). The origin of the decrease in protein expression induction at longer exposures is currently unknown. *E. coli* viability begins to decrease above ~0.1 $J/cm^2$ (at 325 nm light exposure, increasing to ~1 $J/cm^2$ at 390 nm) (*Vermeulen et al., 2008*) and our photon flux was 0.014 and 0.055 $W/cm^2$ for the 342 nm and 390 wavelengths of light, respectively. This means that at exposures above ~10 s, the photon flux in our experiments would begin to affect the cell viability. Our experiments were performed at UV exposures well below lethal doses at these wavelengths, even at the longest exposure times; therefore, we do not expect DNA or RNA damage to cause the exposure dependence shown in *Figure 3b*. We confirmed the cell viability using cells expressing the Were-1-Fluc construct in absence of amino-*t*SS, showing insensitivity to the UV even at multi-second exposures; therefore, the decrease of protein expression at longer exposures likely results from a cellular activity of the amino-*t*SS that is likely unrelated to the Were-1 riboswitch activity. Nonetheless, Were-1–regulated induction of protein expression using submillisecond pulses of light represents a significant advance in the RNA optogenetics field, even if the amplitude of the responses is rather modest.

When testing the same system using amino-*c*SS, bioluminescence decreased in a dose-dependent manner with increasing amino-*c*SS concentration after exposure to 390 nm light (*Figure 3c*). This result strongly suggests that by isomerizing amino-*c*SS to its *trans* isoform with a pulse of light, Were-1 was able to sequester its RBS to inhibit luciferase production. Here, higher amino-*c*SS concentrations resulted in greater inhibition of translation, in line with the proportional conversion of amino-*c*SS to amino-*t*SS (*Figure 3c*). Testing the temporal response of Were-1–regulated Fluc expression in the presence of amino-*c*SS revealed that Were-1 again regulates protein expression optimally at short exposures, showing the highest inhibition at a millisecond of light exposure (*Figure 3d*). Based on the isomerization data (*Figure 1—figure supplement 2*, *Figure 1—figure supplement 3*, *Figure 1—figure supplement 4*), the long-exposure effect is likely due to the ligand reaching a semi-photostationary state combined with an unknown effect on the ligand observed for the longer exposures of the *trans* isoform. No difference in cell density was observed among the experiments, implying that neither the ligand, nor the light pulses affect the bacterial growth, and again suggesting negligible photo-damage to the cells.

We next asked whether Were-1 could reversibly regulate gene expression in vivo, providing an optogenetic tool to reversibly regulate cellular events at the RNA level. Using the same *E. coli*

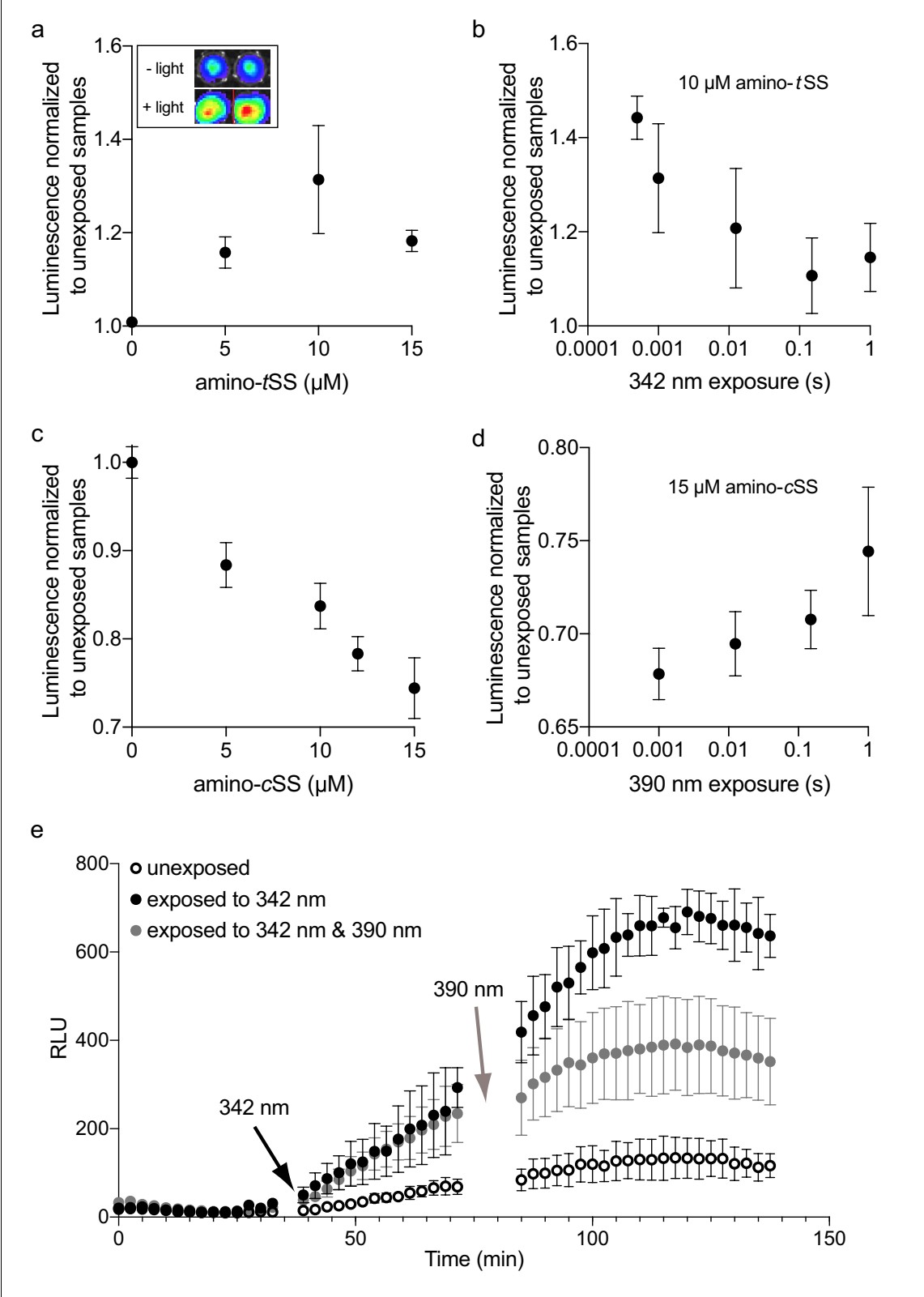

**Figure 3.** Regulation of luciferase expression by the Were-1 photoriboswitch in vivo. (**a**) Normalized amino-*t*SS–dependent bioluminescence (± SEM) of the Were-1-Fluc construct after 1 ms exposure of 342 ± 5 nm light ($\Phi_q = 1.4*10^{-2}$ W/cm$^2$). The largest change in expression was observed in the presence of 10 μM amino-*t*SS. Inset shows the light–dependent bioluminescence of the bacterial cultures at 10 μM ligand. (**b**) Were-1 regulation of luciferase expression (± SEM) in vivo at various exposure times in presence of 10 μM amino-*t*SS. (**c**) Normalized bioluminescence of the Were-1-Fluc E.

*Figure 3 continued on next page*

*Figure 3 continued*

*coli* incubated with amino-*c*SS after 1 s exposure of 390 ± 9 nm light ($\Phi_q$ = 5.5*10$^{-2}$ W/cm$^2$) showing progressively lower protein expression at higher amino-*c*SS concentrations, presumably due to higher production of amino-*t*SS after photoisomerization. (**d**) Change of Fluc expression after photoisomerization of 15 μM amino-*c*SS at 390 ± 9 nm for various exposure times, showing largest photoswitching at 1 ms exposure. (**e**) Regulation of luciferase expression (± SEM) by the Were-1-Fluc construct before exposure, after a 1 ms pulse of 342 ± 5 nm light (black arrow), which resulted in increased bioluminescence (black and gray circles) compared to control (empty circles). Samples exposed to 0.5 ms of 390 ± 9 nm light (gray arrow) showed gradual inhibition of new protein production (gray) compared to samples that were only exposed to 342 nm (black).

The online version of this article includes the following source data and figure supplement(s) for figure 3:

**Source data 1.** Photoswitching of Were-1.
**Figure supplement 1.** Co-transcriptional binding of a ribosome-mimic in vitro under amino-*t*SS photoisomerization conditions.
**Figure supplement 1—source data 1.** Photoswitching of Were-1 binding by the DNA toehold.
**Figure supplement 2.** Were-1 photoregulates translation in vitro.
**Figure supplement 2—source data 1.** Photoregulation of Were-1-regulated translation in vitro.
**Figure supplement 3.** Were-1 photoregulation of protein expression in vivo.
**Figure supplement 3—source data 1.** Wavelength-dependent photoregualtion of Were-1-regulated protein expression in vivo.
**Figure supplement 4.** Lack of photoactivity of the Were-1-Fluc G69C variant.
**Figure supplement 4—source data 1.** Lack of photoswitching of the G69C variant of Were-1.

construct, we measured bioluminescence over two hours in samples that were covered during excitations, and therefore unexposed to light, samples that were exposed to a sub-millisecond of 342 nm light, and samples that were first exposed to 342 nm and later excited at 390 nm for a millisecond. Initial values prior to excitation showed no significant difference in bioluminescence twenty-five minutes post induction (*Figure 3e*). Samples that were then exposed to 342 nm light showed an increase in luciferase expression, compared to the unexposed control. Lastly, bioluminescence of the samples that were subsequently exposed to 390 nm light gradually decreased due to low production of new enzymes post exposure (the existing luciferases presumably continued to yield photons), compared to those that were exposed only to 342 nm, which continued protein production unabated. After ~2.5 hr, samples that were exposed to 342 nm light were 5.5 times more luminescent than unexposed samples and 1.8 times more luminescent than the samples exposed to both 342 nm and 390 nm, suggesting that luciferase expression was regulated reversibly. As an additional control, the G69C mutant was exposed alongside Were-1 and showed no significant difference when G69C excited with 342 nm light, or 342 nm and 390 nm (*Figure 3—figure supplement 4*). These results strongly suggest that Were-1 is a photoriboswitch that can reversibly regulate protein expression in vivo.

## Discussion

Regulation of cellular events with light offers many advantages over other approaches, particularly through ultrafast and spatially resolved deployment of the incident photons in live animals, making optogenetic tools invaluable for the mechanistic dissection of many biological systems (*Fenno et al., 2011*). Most optogenetic tools have been developed based on optical receptors, such as rhodopsins and flavin-binding proteins (*Möglich and Moffat, 2010*), which necessitate their adaptation to the pathways interrogated by the optogenetic tools. Methods based on these receptors are thus limited by the availability of the appropriate protein motifs and their photoactive ligands or prosthetic groups, such as flavin and retinal. Regulatory photoreceptor proteins that bind synthetic ligands have not been, to our knowledge, developed. On the other hand, in vitro selection of RNAs offers a facile approach to identify aptamers that bind a wide variety of molecular targets, including photoactive molecules (*Lotz et al., 2019*). Furthermore, riboswitches, cellular ligand-binding RNAs, have been identified in many bacteria, where they typically regulate transcription and translation, and eukaryotes, where they regulate translation, and RNA splicing and stability (*Breaker, 2011*; *Zhang et al., 2020*). Riboswitches therefore offer an attractive mechanism for the development of synthetic regulatory elements, with expression platforms linked to aptamers binding synthetic ligands, although only a few examples of fully de novo selected synthetic riboswitches exist (*Hallberg et al., 2017*).

We show that novel riboswitches, regulated by synthetic ligands, can be evolved from random libraries fused to expression platforms using in vitro selection. Werewolf-1 is the first synthetic riboswitch that binds only one photoisoform of its ligand and reversibly regulates bacterial protein expression at the RNA level. This work provides a new optogenetic tool for probing RNA-level biomolecular processes in living cells, and serves as a platform to evolve new RNAs of similar function. We expect that Were-1 and similar photoriboswitches will allow reversible photoregulation of a variety of RNA-centered cellular events with a high spatiotemporal resolution in bacteria and multicellular organisms alike, advancing the field of optogenetics. In particular, the fast action of the Were-1 photoriboswitch offers new opportunities for mechanistic studies of translational *cis* regulatory elements, such as riboswitches, in vitro and in vivo. Short pulses of UV light can be deployed using light sources commonly found on fluorescence microscopes coupled to mechanical shutters used in film cameras, which allow exposures down to ~100 μs. Such excitation systems thus allow triggering of photoswitches on timescales shorter than individual steps in RNA and protein synthesis (*Manor et al., 1969*; *Bremer and Yuan, 1968*; *Maaløe and Kjeldgaard, 1966*), and many folding transitions in these macromolecules (*Vogel and Jensen, 1994*; *Young and Bremer, 1976*; *Al-Hashimi and Walter, 2008*; *Crothers, 2001*). In contrast to photocaged ligands, the photoreversibility of the Were-1 photoriboswitch is expected to yield short pulses of translation, allowing finely tuned expression of proteins with the potential of triggering the production of a small number of regulatory proteins in single cells, as well as in individual cells in consortia of microorganisms (biofilms) or complex organisms.

# Materials and methods

**Key resources table**

| Reagent type (species) or resource | Designation | Source or reference | Identifiers | Additional information |
|---|---|---|---|---|
| Strain, strain background (*Escherichia coli*) | BL21(DE3) | Sigma-Aldrich | CMC0016 | Electrocompetent cells |
| Recombinant DNA reagent | pBV-Luc-c36 (plasmid) | This paper | | IPTG inducible Were-1–controlled firefly luciferase gene with T7 RNA polymerase promoter |
| Recombinant DNA reagent | pBV-Luc (plasmid) | Adgene | RRID: Addgene_16539 | Parent plasmid for Were-1 construct |
| Chemical compound, drug | Amino-*t*SS | This paper | | Were-1 ligand, *trans* isoform |
| Chemical compound, drug | Amino-*c*SS | This paper | | *Cis* isoform of the Were-1 ligand |
| Chemical compound, drug | *t*SS | This paper | | Were-1 ligand, *trans* isoform lacking a linker |
| Chemical compound, drug | *t*S | Sigma-Aldrich | Cat# 139939 | *Trans* stilbene |
| Chemical compound, drug | *t*DHS | Santa Cruz Biotechnology | CAS 659-22-3 | 4,4-*trans*-dihydroxystilbene |

## Reagents and equipment

Unless otherwise stated, all reagents were purchased from Sigma-Aldrich. (E)−6'-(2-aminoethoxy)−2,2',3,3' tetrahydro-[1,1'-biindenylidene]−6-ol (amino-*t*SS) was synthesized and prepared as described below. Commercially available reagents were used without further purification. Absorbance spectra were recorded with a Thermo Scientific NanoDrop 1000 spectrophotometer. Fluorescence excitation and emission spectra were measured with a Varian Cary Eclipse fluorescence spectrometer, unless otherwise specified. Bioluminescence was measured using an Andor 866 EMCDD camera, BioTek Synergy H1 plate reader, or IVIS Lumina II.

## Synthesis of (E)−6'-(2-aminoethoxy)−2,2',3,3' tetrahydro-[1,1'-biindenylidene]−6-ol

All starting reagents were commercially available, and of analytical purity, and were used without further treatment. Solvents were dried according to standard methods. $^1$H and spectra were recorded on Varian UNITY INOVA-300 and Bruker Avance-600 instruments. Chemical shifts ($\delta$) are reported in ppm relative to residual solvent peak (DMSO: $\delta_H$ = 2.50 ppm) as internal standard. Accurate mass measurements (HRMS) were obtained by ESI on an Agilent 6530 Q-TOF MS spectrometer. Analytical TLC was performed using a precoated silica gel 60 Å F$_{254}$ plates (0.2 mm thickness) visualized with UV at 254 nm. Preparative column chromatography was carried out using silica gel 60 Å (particle size 0.063–0.200 mm). Purifications by HPLC were performed under the following conditions: Agilent ZORBAX SB-C18 column (5 μL, 9.4 × 150 mm); UV/Vis detection at $\lambda_{obs}$ = 254 nm; flow rate 4 mL/min; gradient elution method H$_2$O (0.1% TFA) – CH$_3$CN (0.1% TFA) from 95:5 to 0:100 in 20 min. Purity of compounds was confirmed using Agilent eclipse plus C18 column (3.5 μL, 4.6 × 100 mm); UV/Vis detection at $\lambda_{obs}$ = 254 nm; flow rate 0.5 mL/min; gradient elution method H$_2$O (0.1% TFA) – CH$_3$CN (0.1% TFA) from 95:5 to 0:100 in 20 min.

### Synthesis details

**Scheme 1.** Synthesis of stiff stilbene derivatives tSS, amino-tSS, and amino-cSS.

**Chemical structure 1.** (E)−2,2',3,3'-tetrahydro-[1,1'-biindenylidene]−6,6'-diol (tSS).

To a stirred stirred suspension of zinc powder (3.20 g, 48.92 mmol) in dry THF (50 ml), TiCl$_4$ (4.67 g, 24.58 mmol) was added over 10 min at 0℃. The resulting slurry was heated at reflux for 1.5 hr. Then a THF solution (50 ml) of indanone **1** (600 mg, 4.05 mmol) was added over 3 hr period by syringe pump to the refluxing mixture. The reflux was continued for 30 min after the addition was complete. After cooling to room temperature, the reaction mixture was poured into a saturated solution of NH$_4$Cl and extracted with CH$_2$Cl$_2$. The organic solutions were dried over MgSO$_4$ and concentrated by rotary evaporation under reduced pressure. The crude product was purified by column chromatography using silica gel (hexane/iPrOH = 10:0.5) to afford **tSS** in 51% yield (273 mg) as a white solid. $^1$H NMR (400 MHz, DMSO-$d_6$): $\delta$ 2.95 (m, 4H), 3.02 (m, 4H), 6.63 (dd, $J$ = 8.1, 2.2 Hz, 2H), 7.02 (d, $J$ = 2.3 Hz, 2H), 7.11 (d, $J$ = 8.0 Hz, 2H), 9.17 (s, 2H). $^{13}$C NMR (100 MHz, DMSO-$d_6$): $\delta$ 29.6, 31.9, 111.0, 114.5, 125.2, 135.0, 137.0, 143.7, 156.1. HRMS (ESI): m/z [M]$^+$ calculated for C$_{18}$H$_{16}$O$_2$ 264.1145; found: 264.1141.

**Chemical structure 2.** (E)−6′-(2-aminoethoxy)−2,2′,3,3′-tetrahydro-[1,1′-biindenylidene]−6-ol (amino-tSS).

A mixture of **tSS** (125 mg, 0.47 mmol), 2-(Boc-amino)ethyl bromide (106 mg, 0.47 mmol), $Cs_2CO_3$ (772 mg, 2.37 mmol) and $nBu_4NBr$ (6 mg, 0.02 mmol) in DMF (15 ml) was heated at 50°C for 6 hr. After cooling to room temperature, $CH_2Cl_2$ was added and the mixture was washed with a saturated solution of $NH_4Cl$, water, brine and dried over $MgSO_4$. The organic solutions were concentrated by rotary evaporation under reduced pressure. The crude product was dissolved in a mixture of $CH_2Cl_2$/TFA (16 ml; 3:1) and stirred at room temperature for 30 min. The reaction mixture was concentrated by rotary evaporation under reduced pressure and purified by HPLC (gradient elution method $H_2O$ (0.1% TFA) – $CH_3CN$ (0.1% TFA) from 95:5 to 0:100) to afford **amino-tSS** (as a TFA salt) in 42% yield (84 mg) as an off-white solid. $^1H$ NMR (400 MHz, DMSO-$d_6$): δ 2.96 (m, 4H), 3.01 (m, 4H), 3.22 (t, J = 5.1 Hz, 2H), 4.17 (t, J = 5.1 Hz, 2H), 6.66 (dd, J = 8.1, 2.2 Hz, 1H), 6.87 (dd, J = 8.2, 2.3 Hz, 1H), 7.03 (d, J = 2.2 Hz, 1H), 7.13 (d, J = 8.1 Hz, 1H), 7.16 (d, J = 2.4 Hz, 1H), 7.27 (d, J = 8.3 Hz, 1H), 8.10 (s, 3H), 9.28 (s, 1H). $^{13}C$ NMR (100 MHz, DMSO-$d_6$): δ 29.58, 29.64, 31.8, 31.9, 38.5, 54.7, 110.8, 111.1, 113.6, 114.8, 125.3, 125.4, 134.5, 135.9, 137.1, 139.6, 143.6, 144.0, 156.2, 156.9. HRMS (ESI): m/z [M+H]$^+$ calculated for $C_{20}H_{21}NO_2$ 308.1645; found: 308.1651.

**Chemical structure 3.** (Z)−6′-(2-aminoethoxy)−2,2′,3,3′-tetrahydro-[1,1′-biindenylidene]−6-ol ((Z)−1; amino-cSS).

A solution of **amino-tSS** (15 mg, 35.6 μmol) in DMSO (1 ml) in NMR cuvette was irradiated with handheld UV lamp (8 W) for 15 min. The resulting mixture of **amino-tSS** and **amino-cSS** was purified by HPLC (gradient elution method $H_2O$ (0.1% TFA) – $CH_3CN$ (0.1% TFA) from 95:5 to 0:100) to afford **amino-cSS** (as a TFA salt) in 20% yield (2 mg) as an off-white solid. $^1H$ NMR (400 MHz, DMSO-$d_6$): δ 2.75 (m, 4H), 2.85 (m, 4H), 3.22 (t, J = 5.1 Hz, 2H), 4.13 (t, J = 5.1 Hz, 2H), 6.63 (dd, J = 8.2, 2.2 Hz, 1H), 6.84 (dd, J = 8.2, 2.2 Hz, 1H), 7.12 (d, J = 8.2 Hz, 1H), 7.25 (d, J = 8.3 Hz, 1H), 7.40 (d, J = 2.0 Hz, 1H), 7.53 (d, J = 2.0 Hz, 1H), 8.02 (s, 3H), 9.25 (s, 1H). $^{13}C$ NMR (100 MHz, DMSO-$d_6$): δ 29.2, 29.3, 34.9, 35.0, 38.5, 64.4, 108.9, 109.4, 114.5, 115.2, 125.8, 125.9, 134.4, 135.7, 138.6, 140.79, 140.84, 141.1, 155.46, 156.1. HRMS (ESI): m/z [M+H]$^+$ calcd for $C_{20}H_{21}NO_2$ 308.1645; found: 308.1644.

## In vitro RNA transcription

RNA was transcribed at 37 °C for one hour in a 50 μL volume containing 40 mM tris-HCl, 6 mM dithiothreitol (DTT), 2 mM spermidine, 1.25 mM each rNTP, 8 mM $MgCl_2$, 1 unit of T7 RNA polymerase, and 5 pmol of DNA template. The transcripts were purified by 10% PAGE under denaturing conditions (7 M urea). RNA was eluted from the gel into 300 μL of 300 mM KCl and precipitated by adding 700 μL of 95% ethanol at −20°C.

## In vitro selection of amino-tSS aptamers

An RNA pool derived from a *B. subtilis* mswA SAM-1 riboswitch, located in the 5′ untranslated region of the *metI* (cystathionine gamma-synthase, also denoted as *yjcI*) gene (**Winkler et al., 2003**;

*Grundy and Henkin, 1998*) was designed by replacing the riboswitch ligand-binding domain with a random region of 45 nucleotides. The anti-terminator stem and upstream half of the transcriptional terminator sequence were partially randomized at a 15% level, and the loop of the terminator stem was fully randomized. The remaining part of the riboswitch, including the downstream half of the transcriptional terminator stem, containing a ribosome binding site (RBS) that binds the 3' end of *B. subtilis* 16S rRNA (3'-UUUCCUCCACUAG-5') (*Band and Henner, 1984*) and an alternative UUG start codon, was retained (*Figure 1—figure supplement 1*). The pool was synthesized by Yale School of Medicine's Keck Oligonucleotide Synthesis facility as a single template strand that was then purified by 10% PAGE and converted into dsDNA by a primer-extension reaction using a primer corresponding to the T7 RNA polymerase promoter. The pool was transcribed at an estimated sequence diversity of $10^{15}$.

## From that pool, RNAs were selected to bind amino-tSS, as follows

PAGE-purified $^{32}$P-labeled RNA transcripts of the pool were precipitated, dried, and resuspended in a solution containing 140 mM KCl, 10 mM NaCl, 10 mM tris-chloride, pH 7.5, and 5 mM MgCl$_2$ (binding buffer). The RNA mixture was heated to 70°C for three minutes and loaded onto agarose beads for a counter-selection step. Binders were discarded and the flow-through was incubated on agarose beads linked to amino-tSS. The beads were shaken for five minutes at room temperature, and the unbound RNA was collected. Amino-tSS beads were then washed with binding buffer for five minutes at room temperature. This washing step was repeated six times. Potential aptamers were then eluted twice with denaturing buffer, consisting of 7 M urea and 5 mM ethylenediaminetetraacetic acid (EDTA) in 45 mM tris, 45 mM borate buffer, pH 8, and heated at 95°C for five minutes. Each fraction was analyzed for radioactivity using a liquid scintillation counter. Elutions were pooled, precipitated, dried, and resuspended in water for reverse transcription. The pool was reverse transcribed, and the cDNA was amplified by PCR and used for the next round of selection.

## Screening of potential amino-tSS binders

After six rounds of in vitro selection, the selected pool was cloned into a TOPO TA plasmid (Invitrogen) and transformed into DH5α *E. coli* cells. Cells were plated on agar containing kanamycin and incubated overnight at 37 °C. Individual colonies were picked from the master plate and inoculated overnight in Luria Broth containing kanamycin. Plasmids were extracted and purified using a Miniprep Kit (QIAGEN), and sequenced (GENEWIZ). Individual clones were PCR-amplified using the library-specific primers and transcribed to test their optical activity in the presence and absence of amino-tSS. tSS emission spectra were collected using an excitation at 355 nm. Were-1 showed the highest increase in amino-tSS fluorescence at 430 nm and was chosen for further analysis.

## Structure probing of Were-1
### T1 nuclease probing

Were-1 RNA was dephosphorylated in a solution of the reaction buffer (50 mM potassium acetate, 20 mM Tris-acetate, 10 mM magnesium acetate, 100 µg/ml BSA, pH 7.9), 1 µg of purified RNA, and 1.5 unit of Shrimp Alkaline Phosphatase (NEB). The reaction was incubated at 37°C for 30 min, and heat-inactivated at 65°C for 5 min.

5'–labeled RNA (8000 cpm) was prepared in reaction buffer (70 mM Tris-HCl, pH 7.6, 10 mM MgCl$_2$, 5 mM DTT) using 1 µg of 5'–dephosphorylated RNA, 2 µCi [γ-$^{32}$P] ATP (Perkin Elmer), and 15 units of T4 PNK enzyme (NEB). The reaction was incubated at 37°C for two hours and PAGE–purified.

The 5'–labeled Were-1 RNA was added into binding buffer and the indicated concentrations of amino-tSS or controls (no ligand, amino-cSS, tS, tDHS, and SAM), and were incubated at 55°C for 5 min and subsequently cooled at room temperature for 5 min. Next, T1 nuclease was added (0.05 units; Thermo Fisher Scientific) and samples were incubated at 37°C for 15 min. All conditions were then quenched with a mixture of 7 M urea and 10 mM EDTA. Afterwards, the RNA was added to an equal volume of phenol:chloroform:isoamyl alcohol (25:24:1) and vortexed. Samples were centrifuged for 3 min at 8,000 RPM, and the aqueous phase was collected and transferred to a new tube.

Samples were fractionated on a 10% PAGE gel and exposed to a phosphor image screen (GE Healthcare) for a minimum of 24 hr. The screen was scanned on a GE Typhoon phosphor imager.

A guanosine-specific sequencing lane was resolved in parallel to all samples using 5′–labeled or 3′–labeled RNA (8000 cpm), as specified, in T1 digestion buffer (250 mM sodium citrate, pH 7) and 0.5 units T1. Reactions were incubated at 55℃ for 5 min and quenched with a solution containing 7 M urea and 10 mM EDTA. RNA was extracted using phenol-chloroform, as noted above. Partial alkaline hydrolysis was also resolved in parallel by adding 5′–labeled or 3′–labeled RNA (8000 cpm), as specified, into a hydrolysis buffer (50 mM NaHCO$_3$, 1 mM EDTA, pH 10). Reactions were incubated at 95℃ for 10 min and quenched in a solution containing 7 M Urea and 10 mM EDTA. RNA was extracted using phenol-chloroform.

### Shape

A selective 2′-hydroxyl acylation and primer extension (SHAPE) reaction, as described (*Spitale et al., 2013*), was carried out on Were-1 in the presence of increasing amino-*t*SS concentrations and 30 µM controls (amino-*c*SS, tDHS, tS, and SAM).

### S1 nuclease probing

Reactions were prepared by adding 3′–labeled Were-1 RNA (8000 cpm) into S1 nuclease buffer (40 mM sodium acetate, pH 4.5, 300 mM NaCl, and 2 mM ZnSO$_4$), and the indicated concentrations of amino-*t*SS, and were incubated at 55℃ for 5 min and subsequently cooled at room temperature for 5 min. Next, S1 nuclease was added (0.2 units; Thermo Fisher Scientific) and samples were incubated at 37℃ for 10 min and quenched in a solution of 7 M Urea and 10 mM EDTA. Samples were extracted using phenol-chloroform and resolved on a denaturing 10% PAGE gel. The gel was then exposed to a phosphor image screen and scanned on a GE Typhoon phosphor imager. The sequences in the degradation pattern were assigned by running T1 digestion and partial alkaline hydrolysis in parallel lanes, as noted above.

### Terbium (III) footprinting

Reactions were prepared by adding 5′–labeled Were-1 RNA (8000 cpm) into the binding buffer, and the indicated concentrations of amino-*t*SS or controls (no ligand, amino-*c*SS, tS, tDHS, and SAM), and were incubated at 55℃ for 5 min and subsequently cooled at room temperature for 5 min. Terbium (III) chloride was added to a final concentration of 10 mM and samples were incubated at 37℃ for 30 min and then quenched with a solution of 7 M urea and 10 mM diethylenetriaminepentaacetic acid (DTPA). Were-1 RNA was extracted using phenol-chloroform, as noted above, and samples were fractionated on a denaturing 10% PAGE gel. The gel was exposed to a phosphor image screen (GE Healthcare) and scanned on a GE Typhoon phosphor imager. The sequences in the degradation pattern were assigned by running TI digestion and partial alkaline hydrolysis in parallel lanes, as noted above.

### In–line probing

Reactions were prepared by adding 3′–labeled RNA (8000 cpm) into the binding buffer, pH 8.5, and the indicated concentrations of controls (no ligand, cSS, tS, tDHS, and SAM). Samples were initially incubated at 55℃ for 5 min and then cooled at room temperature for 5 min, and then incubated at 37℃ for 20 hr. All conditions were quenched in a solution of 7 M Urea and 10 mM EDTA. Were-1 RNA was extracted using phenol-chloroform, as noted above, and run on a denaturing 10% PAGE gel. The gel was exposed to a phosphor image screen (GE Healthcare), and scanned on a GE Typhoon phosphor imager. The sequences in the degradation pattern were assigned by running T1 digestion and alkaline hydrolysis in parallel lanes, as noted above.

All gels were analyzed in ImageJ. Structure predictions of Were-1 in the absence of amino-*t*SS were performed using RNAfold of the Vienna RNA package (*Lorenz et al., 2011*; *Figure 1c*).

### In vitro strand displacement reaction

A dsDNA reporter was designed to contain a toehold that complements the Shine-Dalgarno sequence of the riboswitch, in which the longer (toehold) strand (Rep F) contained the 3′ toehold

sequence, a reverse complement of the Shine-Dalgarno sequence, as well as a 5' fluorescein. The shorter strand (Rep Q) contained a 3' Iowa black quencher (*Supplementary file 1* - Supplementary Table 1). A solution of 2:1 Rep Q:Rep F oligos in binding buffer was incubated at 95°C for 1 min, followed by 25°C for 5 min, to anneal the strands and form the dsDNA reporter construct.

In a Falcon 384-well Optilux Flat Bottom plate, strand displacement was initiated by adding 100 nM of purified Were-1 RNA to 50 nM of toehold-fluorophore reporter. Amino-*t*SS was quickly added to some samples to test for ligand-dependent displacement. Fluorescence emission was recorded in a BioTek Synergy plate reader over a 45 min period under continuous illumination using the following parameters: excitation wavelength, 485 nm; emission wavelength, 520 nm.

### In vitro cotranscriptional toehold-binding kinetics of Were-1

In vitro transcription was performed similarly to the above–described RNA transcription assay with the following modifications: 3 pmol template DNA and 50 nM toehold-fluorophore reporter were used. A 30 µL transcription reaction was initiated by the addition of 4 mM rNTP mix (containing 1 mM of each rNTP) and fluorescence emission of the toehold-fluorophore reporter was recorded in a Varian Cary Eclipse fluorimeter under continuous illumination at 37°C using the following parameters: excitation wavelength, 485 nm; emission wavelength, 520 nm; increment of data point collection, 0.01 s; slit widths, 10 nm. These conditions were used for the entire experiment unless stated otherwise. After an initial fluorescence increase, corresponding to the initial burst of transcription, amino-*t*SS was rapidly added to the solution and fluorescence emission was recorded for 200 s. To switch amino-*t*SS to the *cis* isoform (amino-*c*SS), the solution was excited at 342 nm (slit width, 2.5 nm; $\Phi_q$ = 6.8*10$^{-5}$ W/cm$^2$) for 60 s. Fluorescence emission of the toehold-fluorophore reporter was again recorded for 200 s. To switch the *cis* isoform back to the *trans* state, the solution was excited at 372 nm (slit width, 2.5 nm; $\Phi_q$ = 10*10$^{-5}$ W/cm$^2$) for 60 s. Again, fluorescence emission of the toehold-fluorophore reporter was recorded for 200 s. This process was repeated two to three more times until fluorescence plateaued.

### IC$_{50}$ measurements

A dose-response of the Were-1 riboswitch to the target ligand (amino-*t*SS) was assessed by measuring fluorescence as a function of ligand concentration in the presence of a toehold-fluorophore reporter construct (50 nM). Fluorescence emission was recorded under continuous illumination at 37°C using the following parameters: excitation wavelength, 485 nm; emission wavelength, 520 nm; increment of data point collection, 0.01 s; slit widths, 10 nm. The apparent rate constants were measured and plotted against the amino-*t*SS concentrations (or other ligands, as specified). The data were normalized to the no-amino-*t*SS control. The IC$_{50}$ was extracted from fitting a curve to the graph using the equation:

$$\text{Normalized fluorescence} = 1 - \frac{[\text{ligand}]}{[\text{ligand}] + \text{IC50}}$$

### In vitro co–transcriptional magnesium dependence of Were-1 toehold-binding

Using the same conditions as above, the fluorescence response of toehold-binding to the Were-1 riboswitch in the presence of 8.4 µM amino-*t*SS under various Mg$^{2+}$ concentrations was measured. Fluorescence emission was recorded under continuous illumination at 37°C on a BioTek Synergy H1 plate reader.

### Cloning the Were-1 riboswitch for expression in *E. coli* cells

Were-1 DNA was cloned into the pBV-Luc (Addgene) vector in order to obtain a fused riboswitch-firefly luciferase (Fluc) reporter construct. The PCR primers were designed to add a 5' *EcoRI* site to the template Were-1 DNA upstream of the T7 promoter and a 3' overhang containing 35 nucleotides of the Fluc gene directly downstream of its start codon to replace the Fluc start codon sequence. Both the PCR product and plasmid were digested by *EcoRI HF* and *KasI* (New England BioLabs) and purified. The purified construct was then inserted at the 5' end of the Fluc coding

sequence with T4 DNA ligase (New England BioLabs). The resulting vector was termed Were-1-Fluc (*Supplementary file 1*).

Were-1-Fluc was transformed into DH5α *E. coli* cells and grown overnight on agar plates containing ampicillin at 37°C. Ten colonies were picked from a master plate and individual clones were inoculated overnight in Luria Broth containing ampicillin. Plasmids were purified using a Miniprep Kit (QIAGEN) and individually sequenced (GENEWIZ). Correct constructs were transcribed in vitro and fractionated on an agarose gel to confirm sequencing results by measuring the size of the fused construct. Using the same procedure as above, one clone was analyzed in an in vitro co-transcriptional toehold-binding experiment to test whether the new fused construct was able to function similarly to the stand-alone riboswitch.

## In vitro transcription and translation kinetics

The PURExpress in vitro protein synthesis kit (New England BioLabs) was used to transcribe and translate Were-1-Fluc. Experiments were performed similarly to the kit assay conditions with the following modifications: 200 ng/μL DNA, 100 μM D-luciferin, and 2 mM $MgCl_2$. Amino-$t$SS (or other ligands, as specified) was added in conditions when specified. A control plasmid, pET-Luc2, was also tested in the presence and absence of 11 μM amino-$t$SS. All luminescence data were acquired using an ANDOR camera (EMCCD) at 25°C and analyzed using Solis software, and images were further processed and analyzed using ImageJ.

To test whether Were-1 could regulate luciferase protein expression, samples were prepared under identical conditions and luminescence was measured for approximately 40 mins. Samples were then excited at 342 nm ($\Phi_q = 1.4 \times 10^{-2}$ W/cm$^2$) for 1 s, and luminescence was recorded for approximately 30 mins. Samples were excited at 390 nm ($\Phi_q = 5.5 \times 10^{-2}$ W/cm$^2$) for 1 s to switch Were-1 back to the bound 'off' state, and again, luminescence was measured for approximately 30 mins.

## IC$_{50}$ measurements

A dose-response of the Were-1 riboswitch to amino-$t$SS was assessed by measuring luminescence as a function of increasing target concentration. All data were acquired using an ANDOR camera and analyzed with Solis software, and images were further processed and analyzed using ImageJ, as described above.

## In vivo translation kinetics

Were-1-Fluc was transformed into BL21(DE3) *E. coli* cells and grown overnight in Luria Broth containing ampicillin (OD$_{600}$ = 0.26). 1 mM IPTG was added to each well (containing 45 μL culture) to induce T7 RNA polymerase-driven expression and 100 μM D-luciferin to provide a substrate for Fluc. Amino-$t$SS or amino-$c$SS was also added where specified. Bioluminescence was recorded every five mins for one hour at 37°C using a BioTek Synergy H1 plate reader.

To test whether the Were-1 riboswitch regulates the production of Fluc, samples were prepared under the same conditions. In the presence of amino-$t$SS, bioluminescence was measured on a BioTek Synergy H1 plate reader for approximately 15 mins before samples were excited at 342 nm for 1 s in order to isomerize amino-$t$SS to amino-$c$SS. Luminescence was recorded again for approximately 20 mins. Similar experiments were used regarding amino-$c$SS, with the exception of using 390 nm exposure for 0.5 ms in order to isomerize amino-$c$SS to amino-$t$SS, unless further specified.

## In vivo light exposure analysis

To determine the dependence of the amino-$t$SS exposure on the Were-1-Fluc expression, samples were prepared as described above and loaded into two black-bottom 96-well plates. One plate was used as a control and the other was exposed to 342 nm light ($\Phi_q = 1.4 \times 10^{-2}$ W/cm$^2$) for their specified time using a Nikon FM-10 camera shutter in order to isomerize amino-$t$SS to amino-$c$SS. The same procedure was performed in the presence of amino-$c$SS, except 390 nm light was used ($\Phi_q = 5.5 \times 10^{-2}$ W/cm$^2$) for their specified time to isomerize amino-$c$SS to amino-$t$SS. Bioluminescence was measured on an IVIS Lumina II imaging system 1 hr after light exposure.

To test whether the Were-1 riboswitch regulates the production of Fluc multiple times, samples were prepared as described above and loaded in a black-bottom 96-well plate. The top half of the plate was used as a control, containing the unresponsive G69C mutant, and the bottom half contained Were-1. All samples contained 10 μM amino-$t$SS and had their bioluminescence was measured 25 min after induction on a BioTek Synergy H1 plate reader. Next, the first group of wells remained unexposed to light and the middle and far right samples were exposed to 342 nm ($\Phi_q$ = $1.4*10^{-2}$ W/cm$^2$) for 1 ms. Measurements for all samples were taken 40 min after 342 nm exposure. Finally, the last group of wells (far right) were excited at 390 nm for 0.5 ms. Bioluminescence was measured again, for all samples, 40 min after exposure. The same experiment was repeated with the inactive mutant, G69C, as an additional control. Data were normalized to the unexposed samples, and OD$_{600}$ values were obtained to confirm that there was no cell death from UV damage.

### IC$_{50}$ measurements

A dose-response of the Were-1 riboswitch to the target metabolite (amino-$t$SS) was assessed by measuring bioluminescence inhibition as a function of increasing target concentration in BL21(DE3) *E. coli* cells. Bioluminescence was recorded under continuous conditions at 37°C.

## Acknowledgements

We thank the researchers who provided support for this study: Dalen Chan for synthesizing NAI, and Jennifer Prescher's lab for their luminescence reagents and IVIS setup. This work was supported by grants from the National Science Foundation (1804220 to AL), the National Institutes of Health (5T32GM108561-04 to KA R and 5R01GM094929 to AL), the John Templeton Foundation (through the Foundation for Applied Molecular Evolution to AL), Coordenação de Aperfeiçoamento de Pessoal de Nível Superior (Brazilian Federal Agency for the Support and Evaluation of Graduate Education) - 99999.013571/2013–03 (to LP) and the Czech Science Foundation (GACR 17-25897Y to JM).

## Additional information

### Funding

| Funder | Grant reference number | Author |
| --- | --- | --- |
| National Science Foundation | 1804220 | Andrej Luptak |
| National Institutes of Health | R01GM094929 | Andrej Luptak |
| John Templeton Foundation | | Andrej Luptak |
| Grantová Agentura České Republiky | 17-25897Y | Jiří Míšek |
| National Institutes of Health | Graduate Student Fellowship T32GM108561 | Kelly A Rotstan |
| Coordenação de Aperfeiçoamento de Pessoal de Nível Superior | Graduate Student Fellowship 99999.013571/2013-03 | Luiz FM Passalacqua |
| University of California, Irvine | Miguel Velez Scholarship | Luiz FM Passalacqua |

The funders had no role in study design, data collection and interpretation, or the decision to submit the work for publication.

### Author contributions

Kelly A Rotstan, Resources, Investigation, Visualization, Writing - original draft, Writing - review and editing; Michael M Abdelsayed, Conceptualization, Investigation, Methodology, Writing - review and editing; Luiz FM Passalacqua, Investigation, Methodology, Writing - review and editing; Fabio Chizzolini, A Richard Chamberlin, Methodology; Kasireddy Sudarshan, Resources; Jiří Míšek, Conceptualization, Resources, Supervision, Funding acquisition, Investigation, Methodology, Project

administration, Writing - review and editing; Andrej Luptak, Conceptualization, Supervision, Investigation, Methodology, Project administration, Writing - review and editing

Author ORCIDs

Kelly A Rotstan https://orcid.org/0000-0002-2189-4643
Michael M Abdelsayed https://orcid.org/0000-0002-9622-832X
Luiz FM Passalacqua https://orcid.org/0000-0002-5490-2427
Fabio Chizzolini https://orcid.org/0000-0003-4455-8367
Kasireddy Sudarshan http://orcid.org/0000-0002-7072-3226
A Richard Chamberlin https://orcid.org/0000-0002-4685-4672
Jiří Míšek https://orcid.org/0000-0003-4645-7113
Andrej Luptak https://orcid.org/0000-0002-0632-5442

Decision letter and Author response
Decision letter https://doi.org/10.7554/eLife.51737.sa1
Author response https://doi.org/10.7554/eLife.51737.sa2

## Additional files

### Supplementary files

• Supplementary file 1. Tabulated values shown in graphs of *Figures 1–3* and their figure supplements.

• Transparent reporting form

### Data availability

All data generated or analysed during this study are included in the manuscript and supporting files. Source data file has been provided for all figures in an Excel file.

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
