## [Decision Letter]

**Acceptance summary:**

The manuscript by Rotstan and colleagues describes the selection of a riboswitch that can regulate translation by light. To achieve this, the authors used a SAM-based library to in vitro select for a riboswitch that can selectively bind the trans but not the cis isomer of a stiff-stillbene, a compound that has a unique chemical property in that it can reversibly switch between the two conformations through irradiation at wavelengths 342 nm and 372 nm.

**Decision letter after peer review:**

Thank you for submitting your article "Regulation of mRNA translation by a photoriboswitch" for consideration by *eLife*. Your article has been reviewed by two peer reviewers, and the evaluation has been overseen by a Guest Reviewing Editor and Gisela Storz as the Senior Editor. The reviewers have opted to remain anonymous.

The reviewers have discussed the reviews with one another and the Reviewing Editor has drafted this decision to help you prepare a revised submission.

Fundamentally, both reviewers and I agree that the ability to control riboswitch function by light is an important advance, and that even if the dynamic range of the control is only one log, this could be quite valuable in vitro and in vitro. Thus, we would like to invite you to submit a revised manuscript. Both reviewers and I agree that the data in Figure 3 are poorly presented, and in any case, we are surprised that the dynamic range in vivo is less than the log expected from in vitro characterization. The preferred course of action would be to improve the in vivo experiments so that the full dynamic range can be demonstrated, and for these data to form part of a revised figure or figures. Less preferred, but in principle something we will consider, is to expand the in vitro applications, to show that even a log of dynamic range can be productively exploited.

We are listing below comments collated from both reviewers that cover this and additional points.

1) The writing is too concise and I felt the authors could do a much better job setting up the reader for their results. The Introduction section is extremely short and reads more like a summary. Similarly, the Conclusion/Discussion section needs to be elaborated on. Finally, the Abstract is too fragmented and could be put together in a more cohesive manner.

2) While the in vitro data on the utility of the photoriboswitch in regulating translation is convincing, the in vivo data is not. Figure 3 shows little to no change in luminescence as a function of compound addition and/or light. The change appears to be insignificant. To that end, the axis in Figure 3A needs to be changed, it goes from 0 to 1 abruptly, but then it goes to 1.2. At the end, if the authors claim that this new riboswitch can be used to regulate gene expression, the effects have to be larger. These modest effects on gene expression are nowhere near what studies that used caged ligands reported. These showed light can alter protein levels by orders of magnitude.

3) The presentation corresponding to Figure 3 is unclear. This section and the associated figure should be revised. How were the units on the y axis arrived at? Why not show data before and after the light pulse? How long after the light pulse do the data correspond to? Why was the duration of the light pulse not tested in the in vitro experiments? The effects in Figure 3 seem relatively small compared to the supplementary data for Figure 3, making characterization of the system as "robust" seem inappropriate.

The sensitivity of the system to light exposure time seems to arise all of a sudden. Perhaps this should be raised, considered, and tested in the section describing the in vitro experiments. What is the significant of the pulse time sensitivity for real applications? Will specialized equipment be required?

4) Figure 1C and supplementary figure: It is too difficult to distinguish the dark blue and light blue colors in the secondary structure. It would be helpful if the antiterminator helix were made clearer.

5) Results first paragraph: Is this in the total absence of light? What is the stability of the molecule looks like in typical ambient lighting?

6) The description of chemical and nuclease probing experiments used to measure K_D_ and conformational changes upon ligand binding does not indicate that there were technical replicates of the experiments performed.

7) In general, most figures are lacking keys that are really important to get the information across with clarity.

8) Results paragraph four: please show chemical structures of the controls, tS, tDHS, etc.

9) Results paragraph ten: Use of the term "robust" doesn't seem accurate, especially since that "robust" response trails off rapidly if you expose for even a second and there are significant error bars.

10) Results paragraph ten: This time course in the supplemental is the only in vivo time course presented; it stops at 40 minutes. Since stability of the ligand is examined over 48 hours and the in vitro experiments were over about 3 hours, it would be interesting to determine whether the level of luciferase production stays low over a time course of hours in vivo as well.

---

## [Author Response]

1) The writing is too concise and I felt the authors could do a much better job setting up the reader for their results. The Introduction section is extremely short and reads more like a summary. Similarly, the Conclusion/Discussion section needs to be elaborated on. Finally, the Abstract is too fragmented and could be put together in a more cohesive manner.

We have expanded and rewritten all three sections to improve the flow and overall narrative.

2) While the in vitro data on the utility of the photoriboswitch in regulating translation is convincing, the in vivo data is not. Figure 3 shows little to no change in luminescence as a function of compound addition and/or light. The change appears to be insignificant. To that end, the axis in Figure 3A needs to be changed, it goes from 0 to 1 abruptly, but then it goes to 1.2. At the end, if the authors claim that this new riboswitch can be used to regulate gene expression, the effects have to be larger. These modest effects on gene expression are nowhere near what studies that used caged ligands reported. These showed light can alter protein levels by orders of magnitude.

We have corrected the axis in Figure 3A and changed the labeling of all graphs to make the figure less confusing. Furthermore, we show the kinetics of the double switching experiment and provide a more detailed explanation of the observed data. We hope that the edited figure now more clearly conveys the photoswitching behavior of Were-1.

We wish to address some of the finer points of the manuscript. The riboswitch – the cis-regulatory RNA that binds amino-*t*SS – consistently shows robust translation inhibition in vitro and in vivo. These experiments are presented in Figure 2.We think that this result alone is notable, because there are very few examples of de novo laboratory-evolved (synthetic) riboswitches and no aptamer or riboswitch has been reported for a stilbene. in vivo the riboswitch works in multiple media, including 2YT, LB, and M-9 minimal media enriched with 20% glycerol and ~10% LB, though we have observed, on occasion, some variability in the response curve. Because we have not investigated any effect of the media systematically, we chose not to describe these experiments in the manuscript.

On the other hand, the photoswiching has been modest. We now include an example of kinetics of the toehold binding in vitroafter exposure to short, intense UV light in Figure 3—figure supplement 3B. Under these conditions, the toehold binding change is similar to what we observed in vivo(under the same photon flux Φ_q_ = 1.4*10^-2^ W/cm^2^): at short exposures, photo switching is observed (shown for 0.5 and 4 ms exposures at 10 and 15 µM amino-*t*SS), but at longer exposures, we do not see a significant difference. In vivo we were able to reproduce the data on the pulse length that are shown in Figure 3B (in terms of change in luciferase activity one hour after exposure), but we have not observed a larger induction. Clearly, there is some variability in photoinduction, e.g. when comparing Figure 3, Figure 3—figure supplements 2 and 3, but at this stage we do not understand the origin of these differences.

We expect that the photoswitching presented in Figure 3 may be sufficient to produce a pulse of expression for a regulatory protein, but likely not for a large-scale change in protein production. We also expect that further molecular evolution of the system will yield more robust photoswitches.

To address the bell-shaped curve for the photoactivation of Were-1 as a function of amino-*t*SS concentration (Figure 3A), we now provide a brief discussion of the expected behavior of an incompletely photoisomerized ligand on the riboswitch. Basically, when the photoisomerization is not quantitative (as is typical for this class of molecules), the effective concentration of the ligand changes by a fraction, leading to a shift in the IC_50_ curve, but not its abrogation. This means that at low amino-*t*SS concentrations, photoisomerization is not expected to change the state of Were-1, at intermediate concentrations (near IC_50_), the change is expected to be most pronounced, and at high amino-*t*SS concentrations, partial photoisomerization leads to some conversion to amino-*c*SS, but enough of amino-*t*SS may remain to continue inhibiting the riboswitch. This effect is dramatically different for amino-*c*SS, which typically photoisomerizes to the trans isomer with high efficiency; therefore, photoisomerization of amino-*c*SS at increasingly higher concentrations leads to increasingly higher concentrations of amino-*t*SS and therefore higher inhibition of Were-1. This effect is seen in Figure 3C.

Finally, a comparison with photo caged ligands of riboswitches is somewhat difficult, due to the fact that many of the papers that describe these ligands do not report the photon flux used to illuminate their samples, the known cytotoxicity effect of the photouncaging products are also not investigated, and in most cases the temporal control of gene expression is marginal due to slow kinetics of photouncaging and the use of GFP reporters. For photo reversible compounds, such as stilbene and diazobenzene, the photo conversion is not quantitative. Furthermore, most riboswitches do not change the expression of their downstream genes quantitatively. For example, the SAM riboswitches, from which our expression platform was derived, show variable transcriptional termination in presence of SAM and only partial ribosome binding in vitro (while exhibiting quantitative SAM binding).

3) The presentation corresponding to Figure 3 is unclear. This section and the associated figure should be revised. How were the units on the y axis arrived at? Why not show data before and after the light pulse? How long after the light pulse do the data correspond to? Why was the duration of the light pulse not tested in the in vitro experiments? The effects in Figure 3 seem relatively small compared to the supplementary data for Figure 3, making characterization of the system as "robust" seem inappropriate.

We have edited the figure and the figure legend to better describe the experiments and now include a kinetic run of bioluminescence of bacterial cultures that were not exposed, exposed to 342 nm, and exposed to 342 nm and subsequently to 390 nm light, including the reporter signal before exposure. We also now include an in vitro pulse-length measurement showing changes in toehold binding after ~ms exposure of amino-*t*SS to 340-nm light (Figure 3—figure supplement 1B).

The sensitivity of the system to light exposure time seems to arise all of a sudden. Perhaps this should be raised, considered, and tested in the section describing the in vitro experiments. What is the significant of the pulse time sensitivity for real applications? Will specialized equipment be required?

We now discuss the implications of the temporal control accessible by Were-1 in both the Results section where the experiments are described and in the Discussion and Conclusions section, where we describe that Were-1 is sensitive to pulses shorter (at the reported photon fluxes). The pulses are shorter than the individual steps in RNA and protein synthesis, as well as many folding transitions. We expect that such fast temporal control will allow triggering and monitoring co-transcriptional and translation-initiation steps previously not accessible. Finally, we now mention that no specialized equipment is necessary, other than a UV light source commonly found in fluorescence microscopes (or LED sources now commercially available, such as the AmScope 365 nm gooseneck lamps) and a mechanical shutter on an SLR film camera (which can trigger pulses as short as 1/12000 sec, but more commonly, the lower limit of shutter opening on film cameras is 1/8000 sec). Shorter pulses would require either very fast mechanical shutters, requiring more elaborate optical instrumentation, or pulsed lasers.

4) Figure 1C and supplementary figure: It is too difficult to distinguish the dark blue and light blue colors in the secondary structure. It would be helpful if the antiterminator helix were made clearer.

We have changed the color scheme to better distinguish the individual segments of the secondary structure in Figure 1C and the corresponding starting pool in Figure 1—figure supplement 1.

5) Results first paragraph: Is this in the total absence of light? What is the stability of the molecule looks like in typical ambient lighting?

These stiff stilbene molecules appear to be completely unaffected by visible light. All of our experiments were performed under ambient illumination.

6) The description of chemical and nuclease probing experiments used to measure K_D_ and conformational changes upon ligand binding does not indicate that there were technical replicates of the experiments performed.

While we performed some technical replicate experiments, such as T1 probing, the apparent K_D_ values fell within the range of K_D_s derived from the different types of experiments (such as T1, S1, Tb(III) probing). We therefore opted to report the results from these different methods in an attempt to eliminate any potential bias introduced by individual probing methods.

7) In general, most figures are lacking keys that are really important to get the information across with clarity.

We have now included keys to improve the figure clarity, such as in Figures 2C and 2I, and in Figure 3B, D, and E.

8) Results paragraph four: please show chemical structures of the controls, tS, tDHS, etc.

These are now shown in Figure 2C under the first graph describing their activity.

9) Results paragraph ten: Use of the term "robust" doesn't seem accurate, especially since that "robust" response trails off rapidly if you expose for even a second and there are significant error bars.

For the light-activated expression we changed the term to “modest” since the amplitude is not great, even though the experiment appears “robust” in terms of reproducibility.

10) Results paragraph ten: This time course in the supplemental is the only in vivo time course presented; it stops at 40 minutes. Since stability of the ligand is examined over 48 hours and the in vitro experiments were over about 3 hours, it would be interesting to determine whether the level of luciferase production stays low over a time course of hours in vivo as well.

We now present the kinetic data for in vivo photo switching in Figure 3E, showing that the protein expression peaks at about 2 hrs. As discussed above, we find the ratios of expression for the photo switched experiments somewhat variable (see Figure 3E, Figure 3—figure supplements 2 and 3). As with all protein expression, the effect likely depends on the media, the exact time and cell density of transcription induction, and the time of photoinduction.